# The presence of knockdown resistance mutations reduces male mating competitiveness in the major arbovirus vector, *Aedes aegypti*

Lisa M. Rigby [1,2,3]*, Brian J. Johnson[2], Gordana Rašić[2], Christopher L. Peatey[1], Leon E. Hugo[2], Nigel W. Beebe[4,5], Gunter F. Hartel[6], Gregor J. Devine[2]

1 Australian Defence Force Malaria and Infectious Disease Institute, Gallipoli Barracks, Enoggera, Australia, 2 Mosquito Control Laboratory, QIMR Berghofer Medical Research Institute, Herston, Australia, 3 School of Medicine, University of Queensland, Brisbane, Australia, 4 School of Biological Sciences, University of Queensland, Brisbane, Australia, 5 CSIRO, Brisbane, Australia, 6 Department of Statistics, QIMR Berghofer Medical Research Institute, Herston, Australia

* lisa.rigby@defence.gov.au

## Abstract

### Background

The development of insecticide resistance in mosquitoes can have pleiotropic effects on key behaviours such as mating competition and host-location. Documenting these effects is crucial for understanding the dynamics and costs of insecticide resistance and may give researchers an evidence base for promoting vector control programs that aim to restore or conserve insecticide susceptibility.

### Methods and findings

We evaluated changes in behaviour in a backcrossed strain of *Aedes aegypti*, homozygous for two knockdown resistance (kdr) mutations (V1016G and S989P) isolated in an otherwise fully susceptible genetic background. We compared biting activity, host location behaviours, wing beat frequency (WBF) and mating competition between the backcrossed strain, and the fully susceptible and resistant parental strains from which it was derived. The presence of the homozygous kdr mutations did not have significant effects on blood avidity, the time to locate a host, or WBF in females. There was, however, a significant reduction in mean WBF in males and a significant reduction in estimated male mating success (17.3%), associated with the isolated kdr genotype.

### Conclusions

Our results demonstrate a cost of insecticide resistance associated with an isolated kdr genotype and manifest as a reduction in male mating success. While there was no recorded difference in WBF between the females of our strains, the significant reduction in male WBF recorded in our backcrossed strain might contribute to mate-recognition and mating disruption. These consequences of resistance evolution, especially when combined with other

**Funding:** This study was supported by the Australian Defence Force Malaria and Infectious Disease Institute and the QIMR Berghofer Medical Research Institute. The funders had no role in study design, data collection and analysis, decision to publish, or preparation of the manuscript.

**Competing interests:** The authors have declared that no competing interests exist.

pleiotropic fitness costs that have been previously described, may encourage reversion to susceptibility in the absence of insecticide selection pressures. This offers justification for the implementation of insecticide resistance management strategies based on the rotation or alternation of different insecticide classes in space and time.

## Author summary

The mosquito *Aedes aegypti* is the main vector of dengue, chikungunya, and Zika. Its control relies heavily on the use of insecticides but the rapid evolution of resistance to these chemicals compromises their efficacy. The conservation or restoration of insecticide susceptibility in *Ae. aegypti* populations is therefore of great importance. Insecticide susceptibility can be encouraged if the evolution of resistance is accompanied by fitness costs that favour susceptible mosquitoes in the absence of insecticides. This paper documents the first report of a reduction in mating success directly associated with an isolated mutation that confers insecticide resistance in *Ae. aegypti*. This change in behaviour appears related to alterations in male wing-beat frequency. Our results provide evidence of behavioural changes related to insecticide resistance in *Ae. aegypti*, suggesting a competitive advantage of susceptible individuals in the absence of insecticides in the field.

## Introduction

*Aedes aegypti* Linnaeus (Diptera: Culicidae) is an important vector of mosquito-borne viruses, including dengue, Zika and chikungunya. In the case of dengue, approximately half the world's population is at risk and an estimated 100–400 million new infections are reported each year [1–3]. These diseases are not currently controlled by vaccines or therapeutic drugs and their management is reliant on the control of the vector, using insecticides [4,5]. That approach is challenged by the spread of insecticide resistance, particularly to the pyrethroid class of insecticides [6,7].

Mechanisms of pyrethroid resistance in *Ae. aegypti* include increased metabolic detoxification of insecticides [8,9] and point mutations to the voltage-gated sodium channel (VGSC) [10]. The VGSC is a transmembrane protein ion channel that conducts sodium ions through the plasma membrane of cells in the nervous system. Single amino acid substitutions (point mutations) in these ion channels can alter the shape of the channel and the efficacy with which pyrethroids bind to them. This can protect the mosquito from the depolarising (and lethal) impacts of these insecticides [11]. This resistance mechanism is termed 'knockdown resistance' (kdr) and the functional association of these mutations is well documented. In *Ae. aegypti*, this includes amino acid substitutions at positions 989 and 1016 in the IIS6 region of the VGSC [10,12].

Point mutations in the VGSC gene have previously been associated with pleiotropic effects in *Ae. aegypti* including reduced female body size [13,14], increased larval development times [15], and reduced female fecundity [14]. In other insects, kdr mutations have also been associated with behavioural impacts, including reduced mating success [16–18], olfaction [19,20] and cold tolerance [20]. In mosquitoes, the causal relationship between kdr and its apparent pleiotropies are often equivocal because comparisons are commonly made between a small number of susceptible and resistant strains with poorly described genetic backgrounds. Studies that use backcrossing techniques [15,21] or careful genetic characterisation [18] to verify the

precise impacts of kdr alleles are rare. Understanding and documenting the effects of kdr on mosquito behaviours such as mate-recognition and mating success, may be a key step in understanding how associated pleiotropies may assist remediation strategies aimed at reducing the frequency of resistant alleles and conserving pyrethroid susceptibility [22,23].

In this study, we tested the hypothesis that a double homozygous kdr genotype (V1016G/S989P), isolated by backcrossing in an otherwise fully susceptible genetic background, would affect key mosquito behaviours associated with feeding and mating. We measured wing beat frequency (a primary mate-recognition signal in *Ae. aegypti* [24]) and mating success, demonstrated by the successful transfer of seminal fluid marked with the fluorescent dye, Rhodamine B. We also looked for inter-strain differences in avidity (propensity to blood feed) and time to locate a host by free-flying female mosquitoes.

## Methods

### Ethics statement

Mosquitoes were offered blood from the arm of a human volunteer during blood-feeding experiments. The volunteers for mosquito blood-feeding provided formal written consent under the QIMR Berghofer Human Research Ethics committee, approval: P2273.

### *Aedes aegypti* strains

Two strains of *Ae. aegypti* were used to create an insecticide-resistant backcrossed strain with a double homozygous kdr genotype isolated in an otherwise fully susceptible background. R-TL is a pyrethroid-resistant strain that originated from Dili, Timor-Leste in 2009 [25], and S-Cairns is an insecticide susceptible reference strain of *Ae. aegypti* collected in Cairns, Australia in 2014. The derived backcross (R-BC) carries the homozygous mutations V1016G and S989P from the parental R-TL strain in a susceptible S-Cairns background. These kdr mutations are found in the same domain of the VGSC gene [10] and displayed complete linkage disequilibrium during the backcrossing process [21]. R-BC is resistant to DDT, permethrin, deltamethrin and lambda-cyhalothrin and susceptible to malathion and bendiocarb. The development of the R-BC strain and its genetic and phenotypic characterisation have already been described [21].

### Mosquito rearing

Larval densities, nutrition and environmental conditions were standardised across all strains. Colonies were established and maintained in the QIMR Berghofer Medical Research Institute (QIMRB) insectary at 27 (±1)°C and 75 (±5) % relative humidity (RH), with a photoperiod of 12:12 h light: dark (L: D) cycles, including 30 min crepuscular periods. Eggs were hatched by submerging them in dechlorinated tap water, and resulting larvae were reared at a density of 250 individuals per 3 L of water. Larvae were fed with Tetramin fish food (Tetra, Melle, Germany) *ad libitum* and pupae were transferred to 500 mL containers of water inside 30 x 30 x 30 cm cages (BugDorm, Taichung, Taiwan). Adults emerged and mated freely in these cages and were supplied with 10% w/v sugar water *ad libitum*. They were blood-fed once per week using an artificial membrane feeding system [26] and defibrinated sheep blood (Serum Australis).

### Behavioural characterisation

All measured parameters (avidity, host-location, wing beat frequency and male mating success) were evaluated by comparing the parental strains (S-Cairns and R-TL) to the backcrossed

strain (R-BC) under identical conditions. The key comparison is clearly between R-BC and S-Cairns as their genetic background differs only in that R-BC contains the V1016G/S989P homozygous mutations. Comparisons were conducted in a single controlled temperature room within the QIMRB insectary (conditions described above).

## Male mating success

The dye Rhodamine B (Rho B) was used to label the seminal fluid of male mosquitoes to determine paternity in mating competition assays [27]. One day old male mosquitoes from the S-Cairns, R-TL, and R-BC strains were placed in 30 × 30 × 30 cm cages (BugDorm Store, Taichung, Taiwan) and provided access to 0.1% Rho B (Sigma Aldrich, 95% dye content) (w/v) in 10% sucrose solution for 96 h to label seminal fluid. At the conclusion of the 96 h labelling period, 100% of the Rho B fed males were confirmed for labelling success by visual inspection. A separate group of males from each strain was fed a 10% sucrose only solution as an unstained control group. The Rho B and sucrose solutions were replaced every 48 h. The mating competitiveness of each strain was tested by allowing an equal number of males from two strains (one Rho B marked, one unmarked) to compete freely for females from a single strain. Reciprocal assays (swapping the Rho B stained strains) were performed to ensure that there was no labelling bias (S1 Table). The males and females of each strain were separated as pupae prior to emergence to ensure that all mosquitoes were unmated at the start of the competition assay. For each assay, 20 virgin females (3–5 d old), 10 Rho B marked virgin males, and 10 unmarked virgin males (all 5–6 d old) were placed in a cage (i.e. a 1:1 female: male ratio). Both marked and unmarked males were added to each cage before females were introduced. After 24 h, female mosquitoes were collected and knocked down by freezing at -20˚C for 30 min. Individual females were dissected in 1% PBS solution to isolate the bursa and spermathecae. These were mounted on a slide and gently crushed with a coverslip to break open the spermathecae. Specimens were examined using a fluorescent microscope (Zeiss Axioskopp2) with a fluorescence illuminator (Xcite 120Q) and a Cy3 4040c fluorescence filter (531/40 nm Excitation; 593/40 nm Emission) to determine the presence of Rho B. This filter optimised visual differentiation of Rho B-stained tissues. The presence of Rho B (Fig 1A) indicated the mating success of a marked male, while the presence of an expanded bursa (Fig 1B) in the absence of Rho B indicated a successful mating event with an unmarked male. The absence of sperm within the spermathecae and no expansion of the bursa, indicated that successful mating had

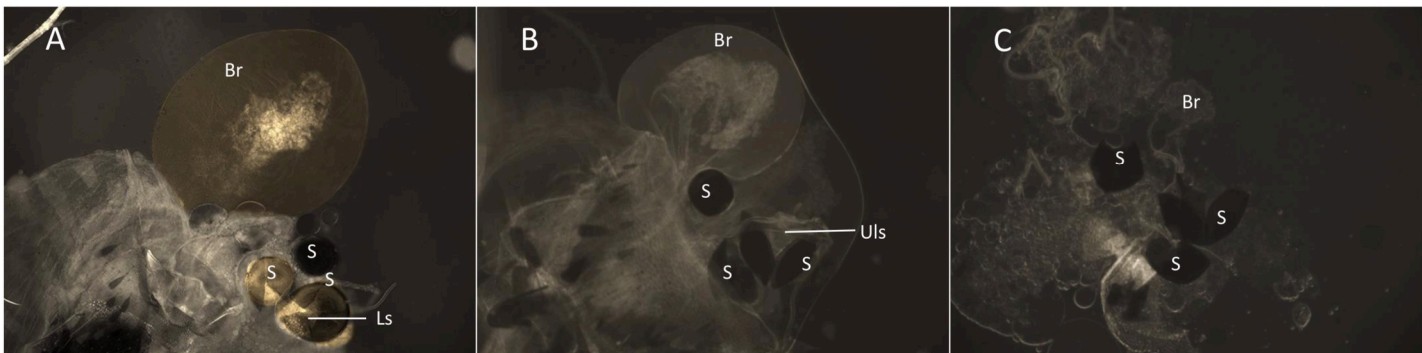

**Fig 1. Examination of female spermathecae for Rho B stained sperm to confirm successful mating by Rho B marked males.** (A) Staining of an expanded bursa (Br), spermathecae (S) and labelled sperm (Ls), indicating a successful mating event with a male marked with Rho B. (B) No staining in the spermathecae, unlabelled sperm (Uls) or bursa but the expanded bursa indicates a successful mating event with a non-Rho B-marked male. (C) Unmated virgin female, no sperm present in the spermathecae and the bursa is not expanded.

not occurred ([Fig 1C]). Three replicates were performed for each treatment. If all females were mated and both competing male groups were equally successful, we would expect 50% of mated females to contain Rho B in the bursa and spermathecae and 50% to show evidence of mating (sperm and an expanded bursa) without staining.

### Free-flight wing beat frequency (WBF) recording

The protocol described by Staunton *et al.*, [28] was used to record the WBF of untethered virgin male and female 3–4 day old, *Ae. aegypti* from each strain. A single male or female mosquito was aspirated from a rearing cage into a vial (60 mL aspirator vial, Bioquip, 2809V, Compton, CA) to record ten seconds of uninterrupted flight (i.e., without landing and resting on the sides of the container). Flight sound was recorded using a TASCAM portable handheld recorder (DR- 22WL; Montebello, California). The sound file was recorded as a high resolution.wav file (32 bit), which was analysed using Audacity software version 2.3.2 (https://www.audacityteam.org). The frequency spectrum of each recording was visualised, and the mean WBF determined from the individual peaks of the fundamental frequency. WBF was recorded for at least 30 male and 30 female individuals from each of the S-Cairns, R-BC and R-TL strains. All recordings were randomised and taken between 0600 and 1800 h.

### Blood-feeding behaviour

Pools of 10 female mosquitoes, 3–4 days old, from the S-Cairns, R-TL and R-BC strains were deprived of sugar solution for 24 h before testing for avidity. Starved females were individually transferred into a rearing cage, allowed to rest for 10 min post-transfer, and offered a blood meal from the forearm of a human adult volunteer for 5 min between 1700 and 1730 h. The number of fully or partially engorged females was recorded. Three batches of 10 mosquitoes from each strain were used for blood-feeding experiments and offered blood from the arm of the same human volunteer.

### Host-locating behaviour

To assess differences in host-location behaviour between the mosquito strains, including the time to locate a human host, "free-flight" assays were used. Pools of 10, 3–4 d old female mosquitoes, deprived of sugar solution for 24 h were released into a white 18 m$^3$ room (2 x 2.8 x 3.2 m) maintained at 26 ± 1°C and 70% RH. Mosquitoes were allowed to acclimatise and rest for 10 min before a human observer entered and sat in the middle of the room (QIMR Berghofer Human Research Ethics approval P2273). Only their lower legs were exposed, ensuring accurate recording of successful landing and feeding events. Each observation was made over 10 min, although most mosquitoes had landed and probed within 3 min. Time to locate the host was recorded for each mosquito and all probing mosquitoes were removed from further observation by aspiration and placed in a holding container. After 10 min, all remaining mosquitoes were captured. Three batches of 10 mosquitoes from each strain were tested. Each test was conducted between 1700 and 1730 h.

### Statistical analyses

We analysed male mating success with respect to male genotype, female genotype and Rho B staining using logistic regression with a log link function to calculate the rate ratio and likelihood ratio estimates with 95% confidence intervals (CI) using JMP Pro version 15.1.0. The differences in time to locate a human host was analysed using one-way ANOVA with Tukey HSD post-hoc analysis. The wing beat frequency data was analysed using two-way ANOVA

with Tukey HSD post-hoc analysis. Data were tested for normality using the Anderson-Darling test. All analyses were undertaken using GraphPad Prism version 7.00 unless otherwise stated. The numerical data used in all figures are included in S1 Data.

## Results

### Male mating success

The logistic regression model showed Rho B marking and the strain of the female mosquito did not have a significant effect on mating success (Table 1). Susceptible S-Cairns males outcompeted the males of the resistant R-BC and R-TL strains for successful mating events with any female strain, while there was no difference in male mating success recorded between the resistant strains. Specifically, the estimate of male mating success for R-BC males in the presence of males from the S-Cairns strain was 32.7% (CI: 27%-38.8%) and this was significantly lower than 50% (i.e. equal mating success between the males of the two strains) (Fig 2A, Table 1). The estimate of mating success for R-TL males in the presence of males from the S-Cairns strain was 40.9% (CI: 34.7%-47.2%) (which was significantly lower than S-Cairns, Fig 2B, Table 1). Finally, when R-BC males competed against R-TL males, the mating success of each strain was not significantly different. The estimate of mating success for R-BC males in the presence of males from the R-TL strain was 48.2% (CI: 41.9%-54.5%) (Fig 2C, Table 1). Significantly lower mating success of males from the resistant strains indicates that the presence of kdr alleles has a pleiotropic effect on the mating success of male *Ae. aegypti*.

### Wing beat frequency

There were significant effects of sex (2-way ANOVA $F_{1,186}$ = 2471, $P < 0.001$) and strain ($F_{2,186}$ = 11.39, $P < 0.01$) on WBF. Males from the R-BC strain displayed a significantly lower WBF in comparison to males from the S-Cairns strain (Tukey's post-hoc test, $P = 0.027$) and males from the R-TL strain ($P < 0.001$, Fig 3A). We recorded no difference in WBF between the S-Cairns and R-TL strains ($P = 0.761$, Fig 3A). There were no statistically significant differences for this trait between females from the three strains (S-Cairns vs R-TL $P = 0.931$; S-Cairns vs R-BC $P = 0.361$; R-TL vs. R-BC strains $P = 0.199$, Fig 3B).

**Table 1. Summary of logistic regression analysis with the log link function for analysis of mating success between males from the S-Cairns and R-BC strains, S-Cairns and R-TL strains, and R-BC and R-TL strains.** The determination of significance for mating success was considered as the deviation from an expected 50% mating success rate. Rate ratios for effect of Rho B staining and female mosquito strain consider a 1:1 ratio.

| Dataset | | |
|---|---|---|
| **S-Cairns vs. R-BC males** | **Effect** | **Estimate (95% CI)** |
| | R-BC Mating Success | 32.7% (27.0%-38.8%) |
| | Rate Ratio Marked | 1.04 (0.86, 1.25) |
| | Rate Ratio Female strain | 0.95 (0.79, 1.14) |
| **S-Cairns vs. R-TL males** | **Effect** | **Estimate (95% CI)** |
| | R-TL Mating Success | 40.9% (34.7%-47.2%) |
| | Rate Ratio Marked | 1.57 (0.91, 1.23) |
| | Rate Ratio Female strain | 1.57 (0.91, 1.23) |
| **R-BC vs. R-TL males** | **Effect** | **Estimate (95% CI)** |
| | R-BC Mating Success | 48.2% (41.9%-54.5%) |
| | Rate Ratio Marked | 0.96 (0.85, 1.1) |
| | Rate Ratio Female strain | 1.07 (0.94, 1.23) |

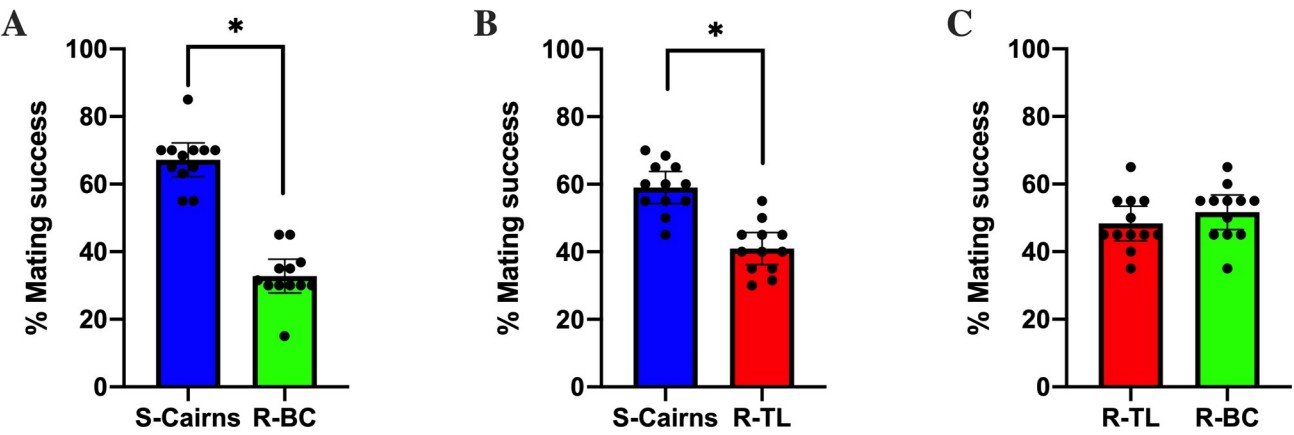

**Fig 2. Percentage mating success.** Mating success between (A) males from the S-Cairns and R-BC strains (B) males from the S-Cairns and R-TL strains (C) males from R-BC and R-TL strains (Mean ± 95% CI). (*: significantly different due to non-overlapping confidence intervals).

### Blood avidity

All starved females from S-Cairns, R-TL and R-BC strains took a blood meal from the human host within 5 min except for one individual from the R-BC strain (S2 Table).

### Host-locating behaviour

All free-flying females from S-Cairns, R-TL and R-BC strains successfully located a human host within ten min. The mean time for R-BC females to find and probe a human host in the free-flight room (113.8 s) did not differ from S-Cairns females (110 s) (Fig 4) ($F_{2,86}$ = 0.9246 $P$ = 0.401).

## Discussion

In the absence of effective therapeutics and vaccines against dengue, chikungunya and Zika, pyrethroid insecticides remain the main tool for managing outbreaks of these diseases by

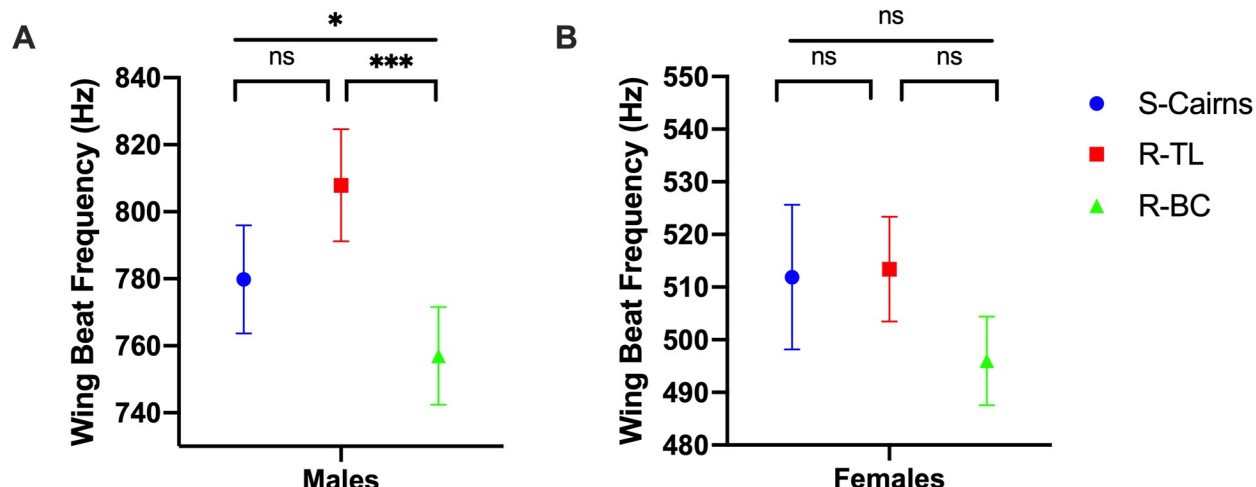

**Fig 3. Wing Beat Frequency.** Average wing beat frequencies recorded from (A) male and (B) female mosquitoes from S-Cairns, R-TL and R-BC strains (Mean ± 95% CI). (*: $P < 0.05$, ***: $P < 0.001$).

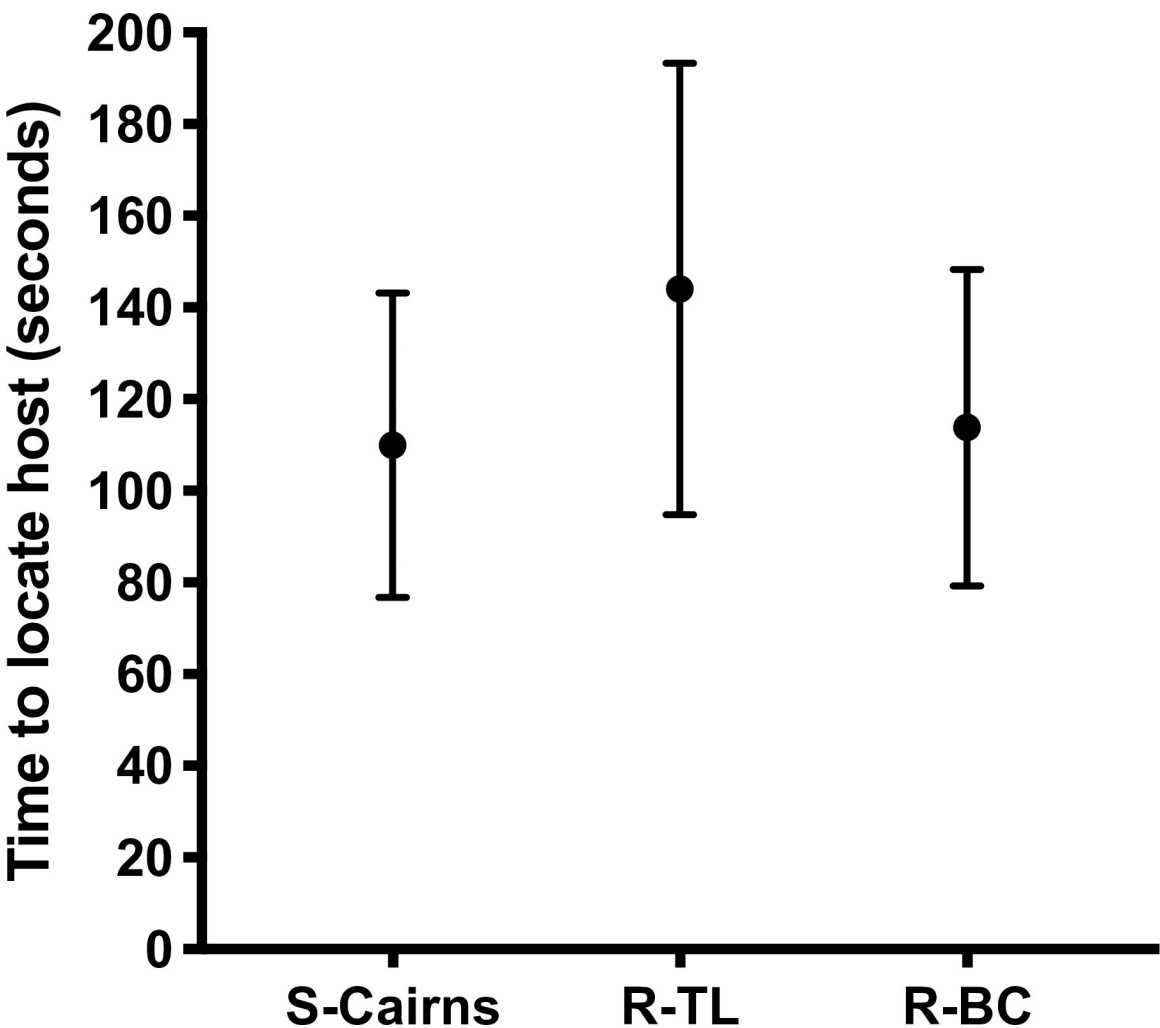

**Fig 4. Host locating behaviour.** Average time (Mean ± 95% CI) for female mosquitoes from S-Cairns, R-TL and R-BC strains to locate and probe a human host.

controlling the primary global vector, *Ae. aegypti*. The effectiveness of that strategy is threatened by resistance to the pyrethroids: a phenomenon that is now extremely widespread across *Ae. aegypti* populations globally [29]. Few insecticide classes have universal approval for dengue control [30,31], so the conservation of susceptibility to the pyrethroids is extremely important. One phenomenon that might assist that process, is the presence of resistance-mediated fitness costs. If resistant strains are less fit than their susceptible counterparts in the absence of insecticides, that difference can form part of the rationale for pursuing resistance management strategies that increase the frequency of susceptible alleles by removing selection pressure. Such strategies include the widely discussed (but rarely implemented) rotation or alternation of different insecticide classes in time or space [32,33].

This paper documents the first report of changes in wing-beat frequency (WBF) and mating success associated with kdr mutations in *Ae. aegypti*. We provide rare empirical evidence of behavioural changes associated with a specific double homozygous kdr genotype, isolated in a susceptible genomic background. Fitness comparisons between insecticide-resistant and susceptible mosquitoes often focus on life-history traits such as survival, development times and

fecundity [13–15] and rarely use strains containing specific resistant genotypes in otherwise susceptible genetic backgrounds [15,21]. Here we used a backcrossed strain with a double homozygous, insecticide-resistance genotype (V1016G/S989P) isolated in a fully susceptible background [21] to explore the pleiotropic effect of these kdr mutations on behavioural traits. The phenomenon that we have documented may affect the frequency of resistant alleles when selection pressure is reduced. We found a reduction in male mating success and a lower male WBF associated with the double homozygous kdr (V1016G/S989P) genotype in the absence of pyrethroids. These are the first carefully controlled observations on the behavioural impacts of isolated kdr mechanisms for *Ae. aegypti*. When considered alongside other reported fitness costs of insecticide resistance including adult survival, body size, fertility, fecundity and larval development times [14,15,21,34] it seems likely that susceptible mosquitoes, in the absence of insecticides, will have a major competitive advantage over resistant forms.

The behavioural correlates of kdr mutations in *Ae. aegypti* in relation to mating competitiveness have been poorly documented [5,22] with the exception of the point mutation conferring dieldrin resistance ('resistance to dieldrin' or rdl) [17,19] and the L1014F kdr mutation [17] that have both been associated with reduced male mating competitiveness in the malaria vector *Anopheles gambiae*. Similarly, dieldrin-resistant *An. stephensi* males (probably carrying 'rdl'), exhibited a reduction in male mating success and subsequent assortative mating by susceptible males [17]. Further, differences in the WBF of male *Drosophila melanogaster* carrying an allele that enhances detoxification of DDT, have been associated with a reduced mating capacity [35].

Mate recognition and pairing in *Ae. aegypti* involves the modulation of male and female wing beat frequencies to achieve "harmonic convergence" [24]. Females may determine whether to mate based on the male's ability to modulate their acoustic signal. Therefore, our observations of a reduced WBF of males from the resistant backcrossed R-BC strain, in comparison to the susceptible parental strain, indicate that pleiotropic effects of kdr on male WBF may have contributed to the reduced mating success of this strain.

In *Ae. aegypti*, there are conflicting studies on the relationship between body size and WBF, reporting either no relationship [36,37], or that smaller mosquitoes have lower WBF frequencies [28]. All recorded WBF for either sex, across all three strains, were within the ranges detailed in previous reports (350–664 Hz for females and 571–832 Hz for males) [36,38]. Males and females of the backcrossed strain (R-BC) are smaller than the susceptible parent strain [21] which may account for the lower WBF observed for males. However, the R-BC strain and the original resistant strain, R-TL, were not significantly different in size and we found no effect of kdr mutations on the WBF of female mosquitoes of the R-BC strain [21]. Although a significant decrease in WBF was noted with smaller R-BC males, there was no general relationship between the smaller body size associated with insecticide-resistant *Ae. aegypti* [14,21] and WBF. This indicates that the modulation of WBF as a pleiotropic effect of the resistance allele may also be dependent on a factor(s) other than body size.

Insecticide resistance-associated decreases in vector-host contact and ingested blood volumes would have significant impacts on pathogen transmission, female survival and egg production. To our knowledge, this is the first investigation that has sought to determine the impact of isolated target-site mutations on short distance host-location and blood-feeding. We did not identify any impact of kdr mutations on female avidity or host location in *Ae. aegypti*, however our experimental design (recording the proportion of females taking a blood meal within 5 minutes and finding and probing a host within 10 min in a laboratory setting) may not have been sufficiently sensitive to identify subtle behavioural impacts. Organophosphate-resistant phenotypes have been associated with reduced avidity and blood ingestion [39] but no such effect has been associated with the kdr genotypes, F1534C [14,15] or V1016G/S989P

[14]. Resistance to deltamethrin and temephos (not associated with kdr mutations), appeared to reduce avidity and the size of the blood meal taken by *Ae. aegypti* [39]. The same resistant strains were also associated with a reduction in the frequency of female insemination by males.

We identified that the insecticide-resistant double homozygous genotype V1016G/S989P reduces male mating success, potentially due to a significant reduction to male wing beat frequency, a key factor affecting female mate recognition. These pleiotropic behavioural impacts, when considered in addition to the more commonly documented effects on survival and fecundity, are likely to have profound implications for the persistence of resistant alleles, the conservation of susceptibility, and the successful immigration of susceptible genotypes in the absence of insecticide pressure. Until novel unresisted chemistries or non-insecticidal disease control methods are available for *Aedes*-borne diseases, such pyrethroid conservation strategies will continue to be one of the few options for sustainable disease management.

## Supporting information

**S1 Table. Male mating competition assay combinations.**
(DOCX)

**S2 Table. The number/percentage of females from each strain successfully taking a human blood meal.**
(DOCX)

**S1 Data. Excel spreadsheet containing, in separate sheets, the underlying numerical data for Figs 2, 3 and 4.**
(XLSX)

## Acknowledgments

This work was done in partial fulfilment of LMR's PhD candidature. LMR was hosted and supervised by the Mosquito Control Laboratory, QIMR Berghofer (GJD). We thank their team for assistance: Igor Filipović, Narayan Gyawali, Oselyne Ong and Melissa Graham. LMR's candidature was registered and supported by the School of Medicine, University of Queensland. We thank Verily Life Sciences, South San Francisco, CA, USA, for the loan of equipment for the measurement of WBF. The opinions expressed herein are those of the authors and do not necessarily reflect those of the Australian Defence Force and/or extant Defence Force Policy.

## Author Contributions

**Conceptualization:** Lisa M. Rigby, Brian J. Johnson.

**Data curation:** Lisa M. Rigby.

**Formal analysis:** Lisa M. Rigby, Gunter F. Hartel.

**Investigation:** Lisa M. Rigby.

**Methodology:** Lisa M. Rigby.

**Resources:** Lisa M. Rigby, Brian J. Johnson.

**Supervision:** Gordana Rašić, Christopher L. Peatey, Leon E. Hugo, Nigel W. Beebe, Gregor J. Devine.

**Writing – original draft:** Lisa M. Rigby.

   Insecticide resistance reduces male mating competitiveness in *Aedes aegypti*

**Writing – review & editing:** Lisa M. Rigby, Brian J. Johnson, Gordana Rašić, Christopher L. Peatey, Leon E. Hugo, Nigel W. Beebe, Gunter F. Hartel, Gregor J. Devine.

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
