## [Decision Letter · Decision Letter 0]

1 Dec 2020

Dear Miss Rigby,

Thank you very much for submitting your manuscript "The presence of knockdown resistance mutations reduces male mating competitiveness in the major arbovirus vector, Aedes aegypti" for consideration at PLOS Neglected Tropical Diseases. As with all papers reviewed by the journal, your manuscript was reviewed by members of the editorial board and by several independent reviewers. The reviewers appreciated the attention to an important topic. Based on the reviews, we are likely to accept this manuscript for publication, providing that you modify the manuscript according to the review recommendations. 

Sincerely,

Felix Hol

Associate Editor

Elvina Viennet

Deputy Editor

Reviewer's Responses to Questions

**Key Review Criteria Required for Acceptance?**

**Methods**

-Are the objectives of the study clearly articulated with a clear testable hypothesis stated?

-Is the study design appropriate to address the stated objectives?

-Is the population clearly described and appropriate for the hypothesis being tested?

-Is the sample size sufficient to ensure adequate power to address the hypothesis being tested?

-Were correct statistical analysis used to support conclusions?

-Are there concerns about ethical or regulatory requirements being met?

Reviewer #1: (No Response)

Reviewer #2: (No Response)

Reviewer #3: all ok

**Results**

-Does the analysis presented match the analysis plan?

-Are the results clearly and completely presented?

-Are the figures (Tables, Images) of sufficient quality for clarity?

Reviewer #1: (No Response)

Reviewer #2: (No Response)

Reviewer #3: all ok

**Conclusions**

-Are the conclusions supported by the data presented?

-Are the limitations of analysis clearly described?

-Do the authors discuss how these data can be helpful to advance our understanding of the topic under study?

-Is public health relevance addressed?

Reviewer #1: (No Response)

Reviewer #2: (No Response)

Reviewer #3: all ok

**Editorial and Data Presentation Modifications?**

Reviewer #1: (No Response)

Reviewer #2: (No Response)

Reviewer #3: (No Response)

**Summary and General Comments**

Reviewer #1: This study was done well and the manuscript was well written. It provides specific evidence enlightening the reader about a cost of insecticide resistance that was missing from the literature. The main questions that arose while I was reading the manuscript (e.g. changes in body size) were addressed in the discussion, which indicates the authors were thorough when considering the strengths, weaknesses, and implications of their work. I have only a few suggestions, and they are all very minor.

Line 128-129: Can the authors please provide the manufacturer’s information for this artificial membrane feeder system or describe how it is made.

Line 188: The interplay between numeric characters in this sentence seems awkward. Consider rephrasing here and in other places to “Pools of 10 female mosquitoes, 3-4 days old, from….”

Line 190-191: Can the authors please clarify whether the same volunteer was used each time or whether different volunteers were used.

Line 317-325: Can the authors please add a brief sentence or two to this discussion regarding whether the change in WBFs observed between their strains match what would be expected based on the differences in body size found in other studies.

Line 364-365: Please add the date this information was cited as per the reference guide for this journal

In there references there is an inconsistent use of upper-case letters in journal titles. Please tidy these up a bit. I believe the journal guidelines ask for sentence case usage.

I have a slight concern about the use of red and green in the same figures as readers who are red-green colourblind may have difficulty with these. Please consider changing one of these colours.

Reviewer #2: Rigby and colleagues have demonstrated a cost of insecticide resistance in Aedes aegypti associated with an isolated kdr genotype, which manifests as a reduction in male mating success. As Aedes control relies heavily on the use of insecticides but the rapid evolution of resistance to these chemicals compromises their efficacy, the conservation or restoration of insecticide susceptibility in Ae. aegypti populations is of great importance. The manuscripts is well written, and the figures and tables are very clear. Great job. 

One thing I was wondering while reading the manuscript was whether females mate with multiple males. If so, how would that have affect the outcome?

Minor comments:

- Omit the p-value from the abstract

- Provide details on the ‘artificial membrane feeding system’

- “ (…) effects that have been previously described, may encourage reversion to susceptibility in the absence of insecticide selection pressures”. Yes, but only if the fitness costs are detrimental.

- Host-location experiment: I assume this falls under the same ethics approval (P2273)

Reviewer #3: The submitted manuscript by Rigby et al reports a reduction in mating success in Ae. aegypti males (R-BC strain) associated with two mutations in the voltage gated sodium channel that confer insecticide resistance. Males of the R-BC strain, that was generated by backcrossing an insecticide resistant strain carrying the two VGSC mutations to an insecticide susceptible strain, also showed a lower wing beat frequency that authors hypothesize could be related to the reduced mating success. However, the underlying causes for the reduction in mating success seem to be more complicated, given also the observation that the parental insecticide resistant strain has a much higher wing beat frequency than the R-BC males, but also a reduced mating success phenotype. Maybe the authors could comment on this a bit more. Overall I found the manuscript well written and it does present some interesting aspects of the potential fitness costs of VGSC mutations.

I have some minor points below. 

Mating success experiment: please add the following information: what was the efficiency of Rho B labelling in males

and what was the percentage of females that had not mated. 

The raw data used to generate the graphs are not provided. I think they should be part of the supplementary information.

Lines 311-313: a reference is needed

Line 314: check the sentence

PLOS authors have the option to publish the peer review history of their article (what does this mean?). If published, this will include your full peer review and any attached files.

Reviewer #1: No

Reviewer #2: No

Reviewer #3: No
---

## [Editor Report · Decision Letter 1]

10 Jan 2021

Dear Miss Rigby,

We are pleased to inform you that your manuscript 'The presence of knockdown resistance mutations reduces male mating competitiveness in the major arbovirus vector, Aedes aegypti' has been provisionally accepted for publication in PLOS Neglected Tropical Diseases.

Best regards,

Felix Hol

Associate Editor

Elvina Viennet

Deputy Editor

Thanks for submitting a revised manuscript and for responding to the concerns raised. I feel the manuscript is ready for publication - congrats on a nice paper!

Felix

---

## [Editor Report · Acceptance letter]

29 Jan 2021

Dear Miss Rigby,

We are delighted to inform you that your manuscript, "The presence of knockdown resistance mutations reduces male mating competitiveness in the major arbovirus vector,* Aedes aegypti*," has been formally accepted for publication in PLOS Neglected Tropical Diseases.

Best regards,

Shaden Kamhawi

co-Editor-in-Chief

Paul Brindley

co-Editor-in-Chief
